# Net Promoter Score (NPS) and Customer Satisfaction: Relationship and Efficient Management

Asier Baquero 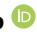

Social Sciences, Law and Business Administration Department, Catholic University of Murcia (UCAM), 30107 Murcia, Spain; abaquero@ucam.edu

**Abstract:** The NPS index is used in the hotel industry to measure customer loyalty and, by extension, customer satisfaction. Many hotel companies set their annual budget based on this index and include it, together with annual economic results, for evaluation when deciding on a potential management bonus. For managers in some companies, achieving a high NPS becomes nearly as important as achieving strong economic results. The purpose of this research is to deepen the study of the NPS index by analysing the existing relationship that the model has with customer satisfaction, focusing on the following main areas of a hotel: reception, cleanliness and room comfort, and gastronomy. To do so, this study uses fuzzy set qualitative comparative analysis (fsQCA). New evidence of value is offered based on the analysis of a sample of six hotels (4 and 5*) located in the Balearic Islands, Spain (Mallorca, Minorca, and Ibiza). In total, 557 surveys were completed in August 2021 and 571 surveys were completed in August 2020, and therefore both sample groups were impacted by a Black Swan (BS) event, the COVID-19 pandemic, in two different stages of its trajectory. The results suggest that in the study sample, the key factor in achieving a high NPS was (1) gastronomy in 2021 (after more than one year of the COVID-19 pandemic), and (2) cleanliness and room comfort in 2020 (at the beginning of the COVID-19 pandemic). These results offer insights for hotel managers, as well as for academics who can develop new lines of research on the subject.

**Keywords:** Net Promoter Score; customer satisfaction; tourism; hotels; Black Swan; COVID-19; fsQCA

## 1. Introduction

Since the exchange of goods and services began, companies have focused on establishing process improvements to offer quality products that meet customer needs. Marketing specifically focuses on strategies to achieve these improvements [1]. During the mid-1990s, the development of technology, computing, and the emergence of the Internet had a significant influence on the market, and data processing is now key to designing marketing strategies [2]. This has led to a commercial projection focused on communication, with a new approach to measuring customer perception in line with the advancement of digital services.

Tourist accommodation is a highly competitive sector. Since the 1990s, demand and expectation for a high-quality service has increased. Presently, the quality of service in the hotel sector goes beyond the facilities themselves, with a specific focus on customer service, customer relations, resolving customer issues, promotions, and facilities access [3].

Wirtz and Lovelock [4] argued that satisfaction can be defined as an attitude judgment after a purchase action or a series of interactions between the consumer and product. Considering this definition, customer satisfaction is the result of the perception of the service received in accordance with the expectation projected by it.

To measure customer satisfaction, it is common for companies to use their own independently prepared questionnaires, in electronic or paper format. The questionnaires are presented to customers and comprise a series of questions in relation to satisfaction with the purchase of a particular product or service. These questionnaires might offer a

closed response option (such as multiple choice or a rating scale), an open response option (allowing customers to include extra comments), or a combination of both [5]. Some companies choose not to use independent questionnaires, preferring to opt for those related to quality models previously formulated and verified in different industries. These include the SERVQUAL or SERVPERF models (which both use approximately 22 questions), or various customer experience metrics such as the Customer Satisfaction Score (CSAT), Customer Effort Score (CES), or the previously mentioned NPS.

NPS was presented by Reichheld in the Harvard Business Review, in an article entitled 'The Only Number You Need to Grow' [6]. The procedure is very simple: customers are asked the single question, 'How likely are you to recommend our company to a friend or colleague?' and are given a response option of a rating scale from 0 to 10. Customers answering 0 to 6 are to be considered 'detractors', 7 and 8 'passive', and 9 and 10 'promoters'. The NPS index, expressed in a percentage, is calculated by subtracting the detractors' scores from the promoters' scores and dividing the total by the number of questionnaires. Passive customers are not considered [7].

With this straightforward, direct, and relatively simple method, Reichheld defended the argument that complex surveys are not necessary and that NPS can replace this method as a more effective way for organizations to determine customer loyalty and even achieve positive financial results [6]. However, some scholars have questioned the reliability of using this 'magic' index as a standalone method [7–11]. Therefore, many companies, from various industries, run a client satisfaction questionnaire with approximately 25 questions, and one of these questions, usually the last one, happens to be NPS. A balance between several detailed questions about the product or service sold together with NPS seems to be the preferred option.

In December 2019, COVID-19 was first detected in Wuhan, China, and the unprecedented outbreak rapidly spread globally. It has since impacted all aspects of society, forcing governments globally to search for and implement solutions [12]. This unprecedented pandemic has harmed both tourism and the global economy [13], and economic forecasts have predicted slowed financial growth and a bleak outlook for populations in countries affected most severely by the COVID-19 outbreak.

According to Taleb [14], a 'Black Swan' (BS) occurrence has three main characteristics: (1) rare, (2) unprecedented, and (3) has a significant impact on society. The COVID-19 pandemic is widely regarded as a BS event. Yet, interestingly, as founder of this theory, Taleb disagrees, presenting the argument that pandemics occur cyclically and that it could have been prevented. Similarly, Yarova Ya et al. [15] asserted that pandemics have existed since the beginning of time, and an extensive list of pandemics can easily be found online. In consensus, pandemics on a large scale such as COVID-19 are extremely uncommon. BS theory is used to classify these high-impact events.

The tourism industry has suffered a significant negative impact as a result of the COVID-19 pandemic. Decisions on travel destinations are heavily influenced by travellers' perceptions of safety and security. Tourists' perceptions of travel risk and management can impact their travel behaviour [16]. Consumer perceptions of danger in relation to travel and management of certain challenges are likely to shift as the disease spreads and uncertainty increases. Typically, tourists are likely to avoid 'dangerous' destinations, where the risk of disruption to travel is much higher. The impact of COVID-19 on tourism destinations has resulted in unanticipated consequences [17].

COVID-19 transmissions and infections have also affected customer loyalty and customer satisfaction within the hospitality industry. There have been instances where tourists have contracted the disease whilst at their holiday destinations [18]. The World Health Organization (WHO) recommended the restriction of movement between countries, advice implemented by the majority of countries globally. As a direct consequence, this change affected customers staying in hotels [19].

The research objective of this study was to propose a model that shows the degree of influence between the satisfaction expressed by clients staying in hotels in the key areas of

reception, cleanliness and room comfort, and gastronomy (dependent variables) and the satisfaction and loyalty expressed and measured by the NPS index (objective variable), in a sample of hotels in the Balearic Islands, Spain. The main investigation focuses on the month of August 2021, with an additional investigation conducted in the same circumstances in August 2020. In this way, it was possible to compare the situation at two different times, both of which were framed within the COVID-19 pandemic. The objective of the study is to understand how hotel managers should ideally approach this issue in order to increase the results in the NPS index. As previously mentioned, there are no known previous publications that have analysed the relationship between specific satisfaction results on areas of hotel services and results in the NPS index. This study makes a detailed contribution to the operation of NPS in the hotel industry that can provide insights to hotel managers and scholars conducting research into this area. In addition, the fsQCA model used in the data analysis presents new information in research on customer satisfaction within the hotel industry.

The main conclusions drawn from the research were the validation of the model, generating successful work options to achieve high levels of NPS, enhancing satisfaction with cleanliness and room comfort for the 2020 sample, and enhancing satisfaction with gastronomy in 2021. The analysis of both the model and the case presented provides novel and high-impact information for scientific studies focused on customer satisfaction, as well as those focused on the NPS index. In addition, it provides empirical insights to hotel owners, managers, and related companies who are interested in improving customer satisfaction and effectively transferring this to the numerical results in NPS.

## 2. Theoretical Background and Development of Propositions

### 2.1. Market Orientation and Quality of Services Related to Customer Satisfaction

The advances in customer communication (predominantly technological and digital) that have occurred since the end of the twentieth century have prompted an evolution from marketing initially defined as marketing 1.0 to the current marketing 4.0. According to Kotler et al. [20], marketing has evolved from being focused on the product (1.0), to focused on the consumer (2.0), to focused on values (3.0), to focused on a social purpose (4.0). This progression has stimulated the implementation of recommendation marketing, whereby customers recommend the brand or company to their personal social network based on their experience. This is amplified in social networks with the use of reviews as a source of brand feedback to measure customer satisfaction [21].

Customers are offered a diverse set of products and services to meet their needs, and their purchase decision is based on expectations. Experiencing a certain level of satisfaction fosters loyalty, with customers then transmitting their experiences and promoting the product or service [1]. The scope of the recommendation is essential for the feedback regarding the brand. Therefore, the ability to adequately measure customer satisfaction is invaluable for strengthening the marketing strategies of companies.

Customer satisfaction is directly associated with the quality of the service. The evaluation of customer satisfaction is a way of obtaining operational information on the quality of the service offered by the company [22]. The dynamization of the market instigated by continuous changes is a result of the competitiveness of organizations to generate value and comply with high quality standards that satisfy the needs of customers. Consequently, this effect has brought with it the prioritization of services, which are focused on health, tourism, and education, amongst other sectors, as an economic factor of global interest [23].

The growth of the services sector is notorious in the case of Spain. After the financial crisis of 2008–2009, a sustained annual increase began from 2013 to 2018, which was dramatically affected in 2020 due to the emergence of the COVID-19 pandemic, according to data from official sources [24,25].

In general, companies must prioritize a solid understanding of customer needs, in order to be able to satisfy them and subsequently guarantee customer loyalty, in such a

way that the quality strategy allows customer satisfaction [26]. This strategy motivates the selection of a suitable instrument to measure customer satisfaction.

A definition of customer satisfaction has been offered by numerous scholars. According to Zeithaml et al. [27], satisfaction can be defined as feelings of pleasure or disappointment that result from comparing the perceived performance of a product or result with expectations. If the performance does not meet expectations, the customer is unsatisfied as a result. If it matches expectations, the customer is satisfied or happy. Similarly, Wirtz and Lovelock [4] (p. 130) state that 'satisfaction is a judgement following a series of consumer product interactions'. Both definitions concur that customer satisfaction is the result of the perception of the service received in accordance with the expectation projected by it.

There are various methods for measuring customer satisfaction. From an operational perspective, some of these methods are simple, whereas others demand greater mathematical complexity in their application. To measure customer loyalty, metrics based on customer satisfaction questionnaires and repurchase intention are mainly used [28]. All methods considered, both implicitly or explicitly, there is a close relationship between service quality and customer satisfaction [29]. Therefore, it is common for companies to use independent questionnaires to measure customer satisfaction or to use pre-settled questionnaires from different quality models [5].

SERVQUAL is a quality model that has been utilized to successfully measure customer satisfaction and establish an evaluative criterion on the quality of the service offered [27]. This concept was developed by Cronin and Taylor [30], who created the SERVPERF model focused on service performance.

Models and indices continued to evolve, with the introduction of the Customer Satisfaction Score (CSAT), consisting of a questionnaire with five Likert scale response options with extreme values and a central value (very satisfied to very dissatisfied), with the percentage satisfaction determined by dividing the sum of the very satisfied and the satisfied by the total number of responses [31].

Another method that is used frequently in market research and regularly in companies is the Customer Effort Score (CES). In this approach, customer interaction with support services or help received by sales teams is evaluated on a scale of 1 to 10. Its aim is to mitigate the amount of effort the client has to exert [32].

In addition to the CSAT and the CES, another instrument that is used very frequently is the NPS, which measures customer recommendation. The purpose of this research was to deepen the study of the NPS index and its relationship with different aspects of customer satisfaction in the main areas of a hotel. No similar research has been found in previous studies, and so this research adds new insights to fill the existing gap between the NPS index and customer satisfaction.

*2.2. Customer Satisfaction and NPS*

The critical factor of consumer loyalty is happiness, a concept particularly relevant to the hospitality industry, which is an industry of happiness [33]. An unhappy customer as a result of a negative experience will have no doubts about taking their business elsewhere, especially given the wealth of options currently at their disposal. The idea of customer satisfaction is complex and comprises several factors.

As previously mentioned, NPS was founded by Reichheld in 2003, in a Harvard Business Review article with the attractive title of 'The Only Number You Need to Grow'. NPS looks at customer loyalty towards a company's products or services after use and is calculated by asking customers a single question—'How likely are you to recommend our company to a friend or colleague?'—with responses allocated on a scale from 0 to 10 [7].

NPS classifies customers according to the answer obtained from the question asked. According to Grisaffe [34], the response is designated on a scale of 0 to 10 points, where 0 indicates that it is extremely unlikely that the brand will be recommended, and 10 indicates that it is very likely that the brand will be recommended. This scale supports the classification of clients into three categories: promoters, passives, and detractors. Promoters

award 9 or 10 points and are clients who had a very positive experience with the company, and it is assumed that they will recommend it and issue positive reviews. Passives award 7 or 8 points and are customers who had a positive experience with the company, but are either not willing to recommend it or are indifferent. They do not usually issue reviews. The third category of customers, detractors, award between 0 and 6 points and are customers who had a negative experience with the company and are willing to comment on it by issuing negative reviews. The method for calculating the NPS index is very simple and easy to interpret. It is calculated by subtracting detractors from promoters—passives are not considered:

$$NPS(\%) = \frac{Promoters - Detractors}{Total \text{ number of respondents}} * 100$$

This percentage index will be defined as follows: $NPS(\%) \leq |100|$. Therefore, it can be both positive and negative. Its interpretation is just as easy to analyse [34].

According to Reichheld and Markey [35], a result of $-100$ to 0 would be unsatisfactory, with $-100$ to $-50$ being deficient, and $-49$ to 0 insufficient. Deficient means that the quality of the service is perceived in a significantly negative way by the client. Insufficient means that the quality of the service is perceived negatively by the customer. A result between 0 and 100 would be satisfactory, with 0 to 49 being sufficient and 50 to 100 excellent. Sufficient is understood to mean that the quality of the service is positively perceived by the customer. Excellent is understood to mean that the quality of the service is perceived in a significantly positive way by the customer.

Reichheld claimed that there was no need for extensive surveys, favouring NPS as a reliable standalone method to determine customer loyalty [6] and consequently achieve organic and sustainable business growth. The reality within the current hospitality sector is that this single question is merged with other more specific questions about the service offered by the particular company to create a satisfaction questionnaire. Management can then obtain an NPS score from the specific NPS question (applying the NPS calculations), in addition to customer satisfaction scores, according to the rest of the questions. An implication of this approach is that NPS and general customer satisfaction indexes can become mixed, distorting the actual value of the NPS and relating it completely to customer satisfaction. Management must be cautious, realistic, and recognize the need to use other tools alongside NPS when assessing the growth rate of a company [11].

There are limitations to the NPS method. For example, a reception manager who is focused on customer satisfaction and obtaining positive feedback about the customer experience might be reproached because of a low NPS score. However, this low NPS might be the result of negative customer experience in other areas of the hotel, such as food quality or noise interference. This superficial analysis can put pressure on departmental managers and negatively affect the ultimate aim of NPS, which is to improve customer satisfaction and loyalty. Due to this, various academics have raised doubts about the efficiency of this single question and the attractiveness of NPS for management [8,9]. Despite this, the success of NPS is to be expected and justified within certain industries, given its potential to offer a very simple and cheap way of obtaining a quantifiable measurement of customer loyalty and satisfaction. It can thus be easily traced, monitored, and used as a tool to apply pressure and regulate management responsibility.

Mandal [11] proved that there are certain industries where NPS is a less reliable indicator when used as a standalone method (an argument recognized even by Reichheld himself), such as database software and cable TV, amongst others.

After investigating six Dutch hospitals, Krol et al. [36] concluded that there is no certainty that NPS alone generates more reliable results for the management of patient satisfaction in comparison with other types of questionnaires. Similarly, following their study on the insurance industry in Denmark, Kristensen and Eskildsen [10] concluded that NPS is a weak predictor of customer loyalty and satisfaction, and that using NPS as the only ratio in business management could lead to misunderstandings in the measurement of these areas.

To predict customer behaviour and satisfaction, companies must adopt a multidisciplinary approach in order to avoid falling victim to a single question metric, which might hinder their final objectives [9,37]. According to Krol et al. [36], considering that the instrument is applied with the same structure to different companies, it facilitates in a simple and operational way the comparison of the quality of the service provided by the company.

### 2.3. COVID-19 Black Swan and Its Impact on the Tourism Industry

The COVID-19 pandemic surprised the world in late 2019/early 2020, causing a global economic and social crisis, with tourism being one of the sectors severely affected [38]. This unstable and uncertain economic environment discourages travel, leading to low demand for tourism [39]. There have been few studies on how to deal with a pandemic in the hotel sector [40], so it is necessary to deepen research that may allow this type of crisis to be adequately managed from all necessary perspectives: financial, operational, and customer satisfaction, amongst others.

The restrictions imposed and the experience of living through a pandemic generate distrust in services on the part of clients, which requires that the areas of interest of client satisfaction be adequately managed in the aftermath of the crisis [41]. With a pandemic, the recommendation system that originates with satisfied customers becomes critically important for accommodation services [42].

In this environment, to achieve customer satisfaction, it is very important to be innovative with services, since restrictions are likely to affect customer perception in one way or another [43]. In addition, given the feasibility of social networks, it is advisable to implement a system of suggestions or collaborative platforms by areas of interest that are part of the measurement of customer satisfaction [44].

The COVID-19 pandemic has been linked in numerous academic studies to the Black Swan (BS) theory. A BS is an event that is out of the ordinary and could have terrible consequences if it happens. BS events are infrequent, have a significant impact, and some argue that they were evident in the past [14].

Some academics, including Taleb, the founder of BS theory, have argued that COVID-19 is not a BS event, since pandemics occur regularly, and that a similar situation occurred some years earlier with SARS. According to Weber [45], it is well-known that coronavirus pandemics have occurred in the past. After a long and anxious wait, the SARS coronavirus outbreak was finally brought under control in 2004. Pandemic influenza and emerging novel diseases were both included in risk registers, though they were not explicitly labelled as such. In essence, it was discovered that a non-flu pandemic was possible.

Any issue affecting medical care, such as a Black Swan event, can lead to disturbances in the world of work, public health, and food supply chains [46]. The social and economic disruption caused by the pandemic is destructive, as it has led to the deaths of millions of people around the world, causing a major impact on society, as BS theory states for its events [15]. Consumers in general expect smooth services without restrictions, and a BS situation may affect these expectations [47,48]. Pandemics are unmanageable due to their suddenness, as has been seen with COVID-19, which clearly relates to BS theory.

The COVID-19 outbreak has undermined the development narratives in place before it. Governments worldwide have enforced the most extreme lockdowns in history to contain the spread of the virus. This disease has harmed many aspects of human existence and industry, including tourism. The pandemic's economic, travel, and social consequences can include long-term health issues for affected persons and loss of friends and family [49].

Strict safety regulations hurt the tourism sector and increase unemployment. Financial, sociological, geopolitical, political, and technological factors influence business today. These changes impact regional or worldwide corporate performance [50]. While wealthy nations have begun vaccination, most underdeveloped nations are still waiting. Patients, doctors, vaccines, and testing facilities for COVID-19 are insecure in many countries [51]. This situation in countries where vaccines are not in general use may affect the world again and slow down the recovery in the tourism industry.

*2.4. Proposals for Research*

After reviewing the empirical evidence on NPS and customer satisfaction, in these pandemic-affected years, this paper aims to demonstrate that:

- Proposition 1: High levels of customer satisfaction increase NPS.
- Proposition 2: High levels of satisfaction measured in key areas such as reception, cleanliness and room comfort, and Gastronomy increase NPS.
- Proposition 3: A key factor that leads to a high NPS can be identified.
- Proposition 3.1: High customer satisfaction with reception is the key factor that leads to a high NPS.
- Proposition 3.2: High customer satisfaction with cleanliness and room comfort is the key factor that leads to a high NPS.
- Proposition 3.3: High customer satisfaction with gastronomy is the key factor that leads to a high NPS.
- Proposition 4: The factor identified as key leading to a high NPS in 2021 should be the same in 2020.

## 3. Materials and Methods

This study analysed two samples: a main sample collected in August 2021, including comments from 557 clients, and an additional comparative sample collected in August 2020, including comments from 571 clients. Both samples were taken from the same 6 hotels (4- and 5-star), which belong to a Spanish chain that manages hotels worldwide. The 6 hotels are located in the Balearic Islands, specifically on the islands of Mallorca, Minorca, and Ibiza.

Clients of the hotels completed a voluntary online questionnaire sent after their stay, having agreed to be contacted at check-in without any type of benefit, in accordance with the usual management procedure of the hotel company.

The customer satisfaction discovery process requires the implementation of some perfect practices in relation to the research for an accurate score to be obtained. The entity running the research questionnaire should pay attention to its design, taking into consideration the research objectives, types of questions, and how it will be responded to, administered, and processed [52]. The online questionnaires used by the hotel company that shared the information for this study were designed by an external professional company and have been in use for several years, sent by email to customers 48 h after check-out.

The data from these surveys were shared by the hotel company in order to facilitate this study. Online questionnaires to measure satisfaction with hotel services have been common practice for many years now, replacing the paper questionnaires traditionally presented in-person during a client's stay. According to Riva et al. [53], online questionnaires are a valid alternative to traditional paper questionnaires if they have been previously validated.

The questionnaire used in the sample under study asked customers about their satisfaction with the service received in various departments of the hotel: the reception, cleanliness and room comfort, gastronomy, entertainment, and the spa. This was assessed using a 5-point response system. If the client rated their satisfaction at between 1 and 3, the system displayed an option to include a personalized comment, qualitatively (qualitive comments were not considered for this study, however). If the client chose 4 or 5, no option was displayed. The final question of the questionnaire asked the client about their intention to return to stay at the hotel and promote it amongst their acquaintances. This question offered a response system from 0 to 10 and corresponded to the NPS.

These online questionnaires were carried out ad hoc by ReviewPro, a Spanish company that is a world leader in applications for the hotel industry, with more than 60,000 clients in 150 countries [54]. According to Hensens [55], ReviewPro is a remarkable example of the software tools currently available on the market to manage customer comments on online sites. This tool facilitates the management of questionnaires made ad hoc for a designated company [56].

In the case of the company under study, customer satisfaction objectives were established based on the results of the online surveys sent to customers. The objective was established at 90% for 5-star hotels, and 86% for 4-star hotels. To obtain the results, the system evaluated the 5-point response system as 1 counting as 1%, 2 counting as 25%, 3 counting as 50%, 4 counting as 75%, and 5 counting as 100%. The result of the NPS is evaluated independently of this departmental satisfaction. NPS is evaluated based on clients' responses on an 11-point scale, as established by the NPS concept. The objective for 5-star hotels is to obtain a result higher than 65%, and for 4-star hotels to obtain a result higher than 45%. The annual objective-based bonus for hotel managers includes the achievement of these satisfaction objectives, for both departmental satisfaction questionnaires and NPS. As indicated by Chen et al. [57], the inclusion of non-financial objectives in the annual bonus for executives intensifies competition amongst them. This is one of the reasons why this study can provide insights to hotel managers, allowing them to better understand the operation of the NPS index. In this way, the aim is to identify combinations of services offered by hotels that lead to an increase in the NPS index of customer satisfaction.

A sample analysis based on fsQCA was used, which is a set theory methodology that considers cases as configurations of causes and conditions, instead of treating each independent variable as analytically distinct and isolated from the rest. This empirical methodology examines the relationships between the outcome of interest (in the case of this study, a high NPS) and all possible combinations (high/low or absent) of its predictors (reception, cleanliness and room comfort, gastronomy). FsQCA summarizes the linguistic 'if-then' rules. The interest in fsQCA is due, fundamentally, to Ragin, who has offered numerous contributions on the subject [58,59]. The main purpose of fsQCA is to fit the data to the theory by going beyond dependence on a single sample, which implies achieving predictive validity [60–63]. This method based on Boolean algebra is suitable for N-samples of small or medium size [64].

The conceptual basis of fsQCA starts from set theory. This allows a detailed analysis of how causal conditions contribute to an outcome. Instead of estimating the effects of individual variables, fsQCA uses Boolean logic to study the relationship between an outcome and all possible combinations of multiple antecedent conditions, allowing researchers to find different combinations of causal variables that suggest different theoretical pathways to a determined result [65]. According to Ragin [59], instead of investigating which factors are more important, fsQCA seeks to know which factors should be combined and in what combinations.

A fuzzy set can be viewed as a continuous variable that has been usefully calibrated to indicate the degree of membership in a well-defined and specified set. Such calibration is possible only using theoretical knowledge, which is essential for the specification of the three cut-off points or qualitative thresholds (full member, non-full member, and maximum ambiguity). For example, cases in the lower ranks of a conventional continuous variable may be totally outside the set-in question, with fuzzy membership scores truncated to 0.0, while cases in higher ranks of this same continuous variable can be totally within the set, with fuzzy membership scores truncated to 1.0.

The measure of consistency, analogous to correlation, is the proportion of cases that are compatible with the result, which is the number of cases that present a certain configuration of attributes and the result, divided by the number of cases that present the same configuration of attributes but do not display the result. This can be seen in the scatter plots. The coverage measure, analogous to the coefficient of determination, assesses the empirical relevance of a consistent subset. This refers to the proportion of the result explained by the model variables—in this case, by the solution. Coverage is based on a causal combination that guarantees that the cases that meet it cover a large enough part of the result to be empirically important. The proportion of members with the result—that is, with high satisfaction—is explained by the solution.

We have a sample of 6 hotels in the Balearic Islands, Spain, for which the NPS index (objective variable) is available, together with customer satisfaction metrics for cleanliness

and room comfort, reception, and gastronomy (dependent variables), calculated as the average of the customer ratings for each hotel. To apply the fsQCA methodology in this research, fsQCA 3.0 (Fuzzy-Set/Qualitative Comparative Analysis Version 3.0) was used. FsQCA methodology analyses have been used successfully in multiple studies in the social sciences [66–70].

The descriptive statistics shown below on the initial data provide a better way to understand the methodology, with each of the dimensions in its original mode (Table 1).

**Table 1.** Descriptive data.

| Dimension | Valid N | Mean | Standard Deviation | Percentile 05 | Median | Percentile 95 |
|---|---|---|---|---|---|---|
| NPS | 6 | 45.73 | 14.17 | 30.71 | 41.81 | 66.36 |
| Reception | 6 | 89.05 | 3.89 | 84.40 | 88.45 | 94.80 |
| Cleanliness and room comfort | 6 | 82.83 | 6.27 | 76.00 | 82.20 | 91.70 |
| Gastronomy | 6 | 81.92 | 2.16 | 79.80 | 81.50 | 85.50 |

Both the analysis of the model and the analysis of the specific case presented add novelty to scientific studies focused on customer satisfaction and in-depth insights into the NPS index. They also provide empirical insight to hotel owners, managers, and those in similar consumer industries.

## 4. Results

Prior to the realization of the fuzzy model, calibration was carried out, which in this research indicates the extent to which hotels can be considered members of groups that vary according to the NPS index, and the services offered.

In this investigation, four factors were analysed (Table 2).

**Table 2.** Factors to be analysed.

| Condition/Outcome | | Code | Items |
|---|---|---|---|
| Outcome | NPS index | NPS | Degree of customer satisfaction with their stay at the hotel expressed as a recommendation to third parties. |
| Antecedent conditions | Satisfaction with reception | Reception | Degree of customer satisfaction with the service in the reception area (check-in, etc.). |
| | Satisfaction with cleanliness and room comfort | Clean_conf_room | Degree of customer satisfaction with cleanliness and room comfort. |
| | Satisfaction with gastronomy | Gastronomy | Degree of customer satisfaction with the service and quality of food offered for breakfast, lunch, dinner, and room service. |

Each of these quantitative variables was calibrated to grant degrees of membership or belonging to previously defined groups. The details are given below, where the first value corresponds to the calibrated value 1 (full member), the middle corresponds to the ambiguous calibrated value 0.5, and the third value corresponds to the calibrated value 0 (non-member):

nps_fz = calibrate (nps, 66.36, 41.81, 30.71).
recep_fz = calibrate (reception, 94.80, 88.45, 84.40).
hcflp_fz = calibrate (h_confclean, 91.70, 82.20, 76.00).
gastro_fz = calibrate (gastronomy, 85.50, 81.50, 79.80).

These sets are shown in Table 3.

**Table 3.** Calibration factors.

| Conditions | | Set Membership |
|---|---|---|
| Outcome | NPS index | Hotels whose customers have a high NPS. |
| Antecedent conditions | Satisfaction with reception | Hotels whose customers are highly satisfied with reception. |
| | Satisfaction with cleanliness and room comfort | Hotels whose customers are highly satisfied with cleanliness and room comfort. |
| | Satisfaction with gastronomy | Hotels whose customers are highly satisfied with gastronomy. |

*4.1. Sufficiency Analysis*

Once the results and all the conditions had been calibrated ('fz' indicates a calibrated variable), this study proceeded to extract the 'Truth Table', in which all the possible configurations are listed. According to Ragin [59], there are $2^k$ configurations or rows, where k is the number of conditions (in this case $2^3$ = 8 combinations). The value 1 in each setting indicates a calibrated variable score greater than or equal to 0.5 (that is, closer to the full member category) and 0 indicates calibrated variable values less than 0.5 (closer to the non-member category). They are ordered from the highest to the lowest number of cases with a membership score greater than 0.5 in that configuration (column 'number' is the accumulated %), and the consistency of each of them is shown based on the subset relationship with the result, as shown in Table 4.

**Table 4.** Truth Table.

| recep_fz | hcflp_fz | gastro_fz | number | nps_fz | Raw Consist. |
|---|---|---|---|---|---|
| 1 | 1 | 1 | 1 (16%) | | 0.992308 |
| 1 | 1 | 0 | 1 (33%) | | 0.989362 |
| 1 | 0 | 1 | 1 (50%) | | 0.419643 |
| 0 | 1 | 0 | 1 (66%) | | 0.764286 |
| 0 | 0 | 1 | 1 (83%) | | 0.892857 |
| 0 | 0 | 0 | 1 (100%) | | 0.524096 |
| 1 | 0 | 0 | 0 (100%) | | |
| 0 | 1 | 1 | 0 (100%) | | |

The next step is the selection of a consistency threshold to distinguish causal combinations that are subsets of the result from those that are not. Values below 0.80 in this column indicate substantial inconsistency [59]. This study selected 0.893 as the consistency threshold and assigned the value 1 to the result variable (nps_fz) when the consistency of that configuration exceeded the 0.893 threshold. Conversely, when it was below the threshold, it was set as 0, as shown in Table 5.

**Table 5.** fsQCA output.

| recep_fz | hcflp_fz | gastro_fz | Number | nps_fz | Raw Consist. |
|---|---|---|---|---|---|
| 1 | 1 | 1 | 1 | 1 | 0.992308 |
| 1 | 1 | 0 | 1 | 1 | 0.989362 |
| 1 | 0 | 1 | 1 | 0 | 0.419643 |
| 0 | 1 | 0 | 1 | 0 | 0.764286 |
| 0 | 0 | 1 | 1 | 1 | 0.892857 |
| 0 | 0 | 0 | 1 | 0 | 0.524096 |

The resulting intermediate solution is two combinations that sufficiently increase the customer's NPS index. We view complex and parsimonious solutions as the two extremes of a single complexity/parsimony continuum. Any solution that is a subset of the most parsimonious solution and a superset of the most complex solution is a valid Truth Table solution [71]. As shown in Table 6, these intermediate solutions use only a subset of the assumptions that were simplified and that are used in the more parsimonious solution.

**Table 6.** fsQCA output. Intermediate solution (reduced final set): leading to high NPS.

| Sets | Raw Coverage | Unique Coverage | Consistency |
|---|---|---|---|
| hcflp_fz*recep_fz | 0.633 | 0.498 | 0.995 |
| gastro_fz*~hcflp_fz*~recep_fz | 0.346 | 0.211 | 0.892 |
| Solution coverage: 0.844 Solution consistency: 0.953 | | | |

The final solution can be expressed as follows:

hcflp_fz*recep_fz + gastrocfz*~hcflp_fz*~recep_fz

In fuzzy set theory, there are three common operations that are used in the present investigation. These are negation (represented by ~), intersection (logical AND, represented by *), and union (logical OR, represented by +):

Logical negation (~): (membership in the set ~ M) = 1.0—(membership in the set M).

Logical AND (*) is carried out by taking the minimum membership score of each case in the sets that are combined.

Logical OR (+) is carried out by taking the maximum membership score of each case in the sets that are combined.

The two combinations shown above in Table 6 sufficiently increase NPS in 95.3% of the cases and cover 84.4% of the cases. Both the combination of high satisfaction with the quality of cleanliness and room comfort and high satisfaction with reception and the combination of high satisfaction with gastronomy and low satisfaction with cleanliness and room comfort and reception lead to a sufficient increase in NPS. These attributes are in line with the statements of Dolnicar and Otter [72] about the important areas of a hotel to be measured in satisfaction surveys.

In view of the results, it seems clear that the category of the hotel (four- or five-star) and the segment to which the hotel belongs (five-star luxury hotel, four-star adults-only hotel, four-star family resort) decisively influences the fulfilment of the conditions and is consistent or inconsistent with the result. The consistent ones are the hotels whose second coordinate value is greater than or equal to the first, and the relevant consistencies are those where the first coordinate value is greater than 0.5 (therefore, both are greater than 0.5). The first value is the degree of belonging to the combination and the second is the degree of belonging to the result (high NPS). The inconsistent ones are those where the first value is greater than the second, and the seriously inconsistent ones are those where the second is greater than 0.5 (therefore, both are greater than 0.5).

In the model presented in this study, there are no serious inconsistencies:

hcflp_fz*recep_fz: 4* adults-only hotel #1 (0.95,0.95), 4* adults-only hotel #2 (0.59,0.62).

gastro_fz*~hcflp_fz*~recep_fz: 5* hotel (0.61,0.89).

In hotels that meet the first condition (four-star adults-only hotel #1, four-star adults-only hotel #2), the gastronomy service is not a determining factor in customer satisfaction if the reception and cleanliness and room comfort are rated highly. In hotels that meet the second condition (five-star hotel), gastronomy is relevant because, even with poor reception and cleanliness and room comfort, NPS will be high if gastronomy is good.

Figure 1 shows the consistency and coverage of the solution, in a scatter plot that compares the solution with the result: a combination that systematically has all the (calibrated) scores less than or equal to the result scores (upper triangle) is said to be a subset

of the result and the consistency is high. Hotels that are below the diagonal indicator are inconsistent with the result, and those that are above it are consistent. Note that there are no cases in the red zone, meaning there are no serious inconsistencies.

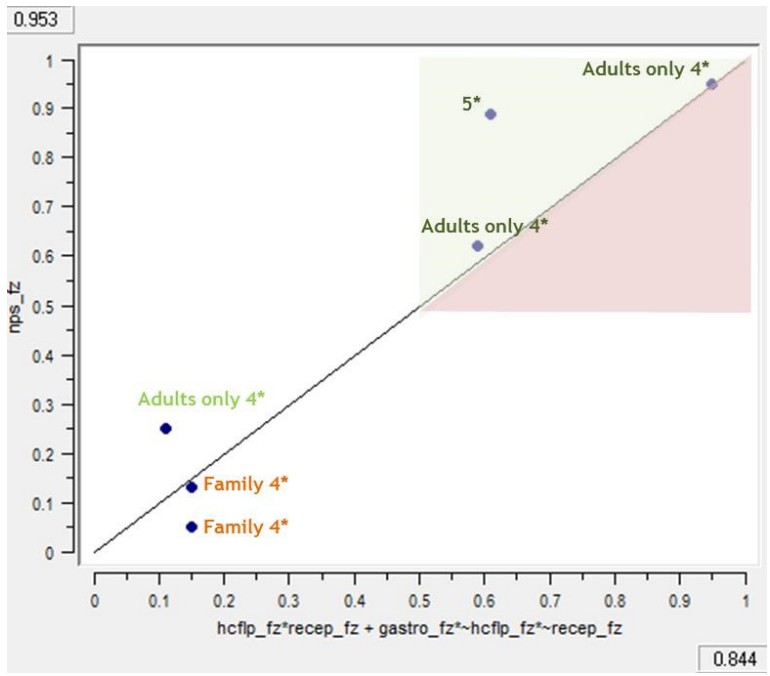

**Figure 1.** Plot of "nps_fz" against "hcflp_fz * recep_fz + gastro_fz * ~hcflp_fz * ~recep_fz".

According to Ragin [58], when analysing sufficiency, we compare the membership scores of the outcome not only with the score of each individual case, but also with the scores of all possible causal expressions. If all cases are above the diagonal indicator, this indicates that the membership scores of the outcome are consistently greater than the membership scores of the causal combination. The causal expression is therefore a subset of the outcome, which is the set-theoretic way to express sufficiency.

The green zone represents the most consistent cases with the solution, and the red zone represents the most inconsistent cases with the solution. Within each group, there are degrees of relevance depending on whether the score for belonging to the combination is less than or greater than 0.5 (upper right quadrant). An inconsistency in the red triangle is more serious ($Xi \geq 0.5$, $Xi > Yi$), and a consistency in the green triangle is more relevant ($Xi \leq 0.5$, $Xi \leq Yi$).

Regarding the most relevant consistencies in the global solution, there is the five-star hotel and two four-star adults-only hotels. Note that the two four-star family hotels are inconsistent with the global solution, in addition to both having very low satisfaction both in the combinations of parameters and in the overall NPS (they would be the worst-rated hotels).

As the fsQCA technique is not symmetrical, unlike other quantitative estimation techniques, it is advisable to look at which combinations of factors lead to a low NPS, since a result does not always explain its denial. The resulting configuration for the negative result is shown in Table 7.

With a coverage of 83.9% and a consistency of 89.1%, both the combination of low satisfaction in gastronomy and reception and the combination of high satisfaction in gastronomy and reception and low satisfaction in cleanliness and room comfort lead sufficiently to a low NPS. The first solution means that, in the presence of bad gastronomy and bad reception, cleanliness and room comfort is not decisive for NPS dissatisfaction. In the second solution, it is the opposite: if cleanliness and room comfort is bad, NPS will be low, even if reception and gastronomy are good.

**Table 7.** fsQCA output. Intermediate solution (reduced final set): leading to low NPS.

| Sets | Raw Coverage | Unique Coverage | Consistency |
|---|---|---|---|
| ~gastro_fz*~recep_fz* | 0.669 | 0.488 | 0.867 |
| gastro_fz*~hcflp_fz*recep_fz | 0.350 | 0.170 | 0.973 |
| Solution coverage: 0.839 | | | |
| Solution consistency: 0.891 | | | |

*4.2. Necessity Analysis*

When analysing necessity, researchers must look for causal conditions with membership scores that are consistently greater than outcome membership scores. If there is a causal condition where this happens in all cases, then this condition passes the test of necessity and the outcome is therefore a subset of the causal condition, which is the set-theoretic way to express necessity [73].

According to Table 8, the hcflp_fz condition, cleanliness and room comfort, is the one that is closest to the necessary condition, since its consistency is 0.799, close to 0.80. In the sufficiency analysis, it was foreseen that, to a certain extent, the quality of cleanliness and room comfort was a minimum condition and was therefore necessary for high satisfaction, since it appeared in both solutions. Therefore, it can be concluded that cleanliness and room comfort tends to be a necessary condition to obtain high satisfaction.

**Table 8.** Necessity analysis.

| Necessity Analysis | Consistency | Coverage |
|---|---|---|
| gastro_fz | 0.702 | 0.757 |
| ~gastro_fz | 0.470 | 0.409 |
| hcflp_fz | 0.799 | 0.799 |
| ~hcflp_fz | 0.422 | 0.392 |
| recep_fz | 0.661 | 0.649 |
| ~recep_fz | 0.605 | 0.571 |

As demonstrated in Figure 2, half of the hotels are below the diagonal indicator, which is an essential condition for meeting customers' needs [73]. Therefore, two-dimensional scatterplots that represent the arithmetic relationship between the two conditions (result and predictor) help to visualize and understand this concept of a subset (the result is a subset of the condition).

Note the important difference between the application of the subset principle to the assessment of sufficiency and its applications to the assessment of necessity. To demonstrate necessity, the researcher must show that the outcome is a subset of the cause. To support an argument of sufficiency, the researcher must demonstrate that the cause is a subset of the outcome.

*4.3. Comparative Study with 2020*

If we apply the same model for the same hotels in 2020, we obtain the results shown in Table 9.

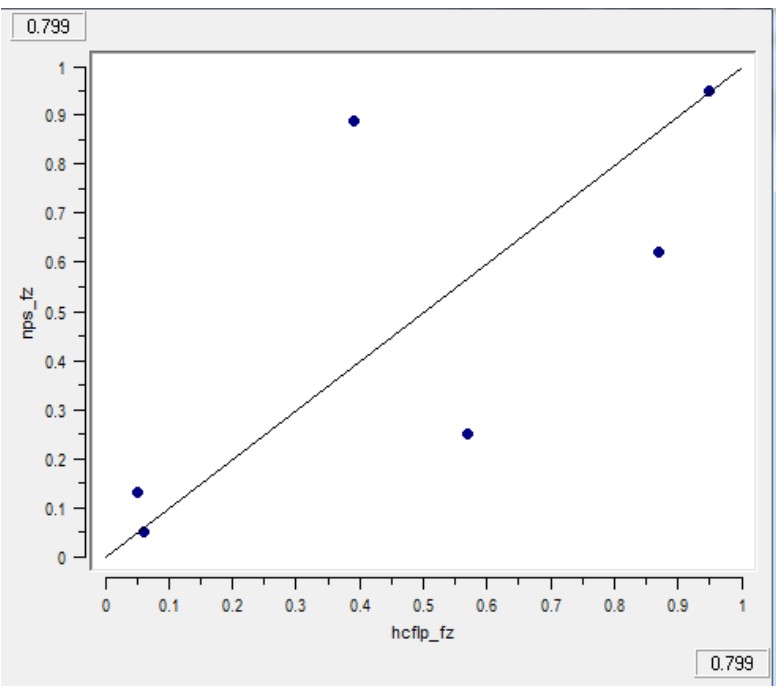

**Figure 2.** Plot of "nps_fz" against "hcflp_fz".

**Table 9.** fsQCA output. Intermediate solution (reduced final set): leading to high NPS in 2020.

| Sets | Raw Coverage | Unique Coverage | Consistency |
|---|---|---|---|
| ~gastro_fz*hcflp_fz*~recep_fz | 0.450 | 0.184 | 0.992 |
| gastro_fz*hcflp_fz*recep_fz | 0.680 | 0.414 | 0.960 |
| Solution coverage: 0.865 | | | |
| Solution consistency: 0.964 | | | |

Ragin asserts that 'to support an argument of sufficiency, the researcher must demonstrate that the cause is a subset of the outcome' [74] (p. 49). In this way, this study observes that cleanliness and room comfort is key in 2020. This condition, in addition, is necessary to achieve a high NPS. Following the same logic, it also has a central influence on achieving a low NPS. In the presence of poor cleanliness and room comfort, the NPS will be low no matter how the other services are rated.

As can be seen in Figure 3, there is a group of hotels with poor Gastronomy and Reception, and a group where these two areas are both highly rated. In both groups, if Cleanliness and Room Comfort is good, NPS will be high.

### 4.4. Differences between 2021 and 2020

In 2020, at the beginning of the COVID-19 pandemic, the key factor is cleanliness and room comfort. If this factor is positive, NPS will be high: four-star adults-only hotel #3 has poor reception and gastronomy, but its cleanliness and room comfort is rated highly. In the 2021 study, the key factor is gastronomy. There are hotels with poor cleanliness and room comfort and poor reception satisfaction but good gastronomy that are highly rated on the NPS (five-star hotel).

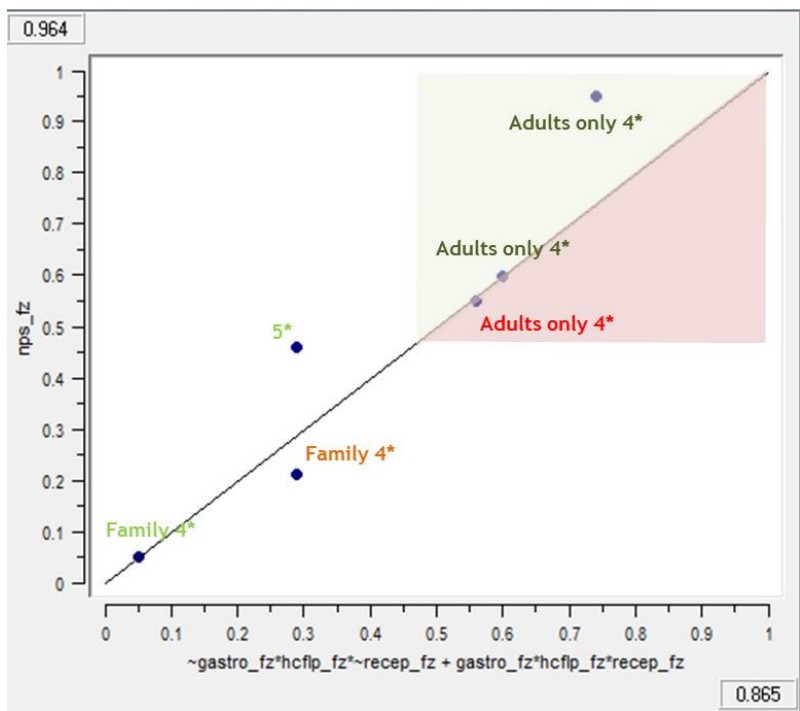

**Figure 3.** Plot of "nps_fz" against "~gastro_fz*hcflp_fz*~recep_fz + gastro_fz*hcflp_fz*recep_fz".

## 5. Conclusions, Limitations, and Future Research

The purpose of this research was to deepen the study of the NPS index and its relationship with different aspects of customer satisfaction within the hotel industry.

Currently, NPS is used by numerous hotel companies as a main indicator to measure customer loyalty and satisfaction, as well as a key element in the consideration of an annual bonus for hotel management. Any progress made in clarifying this operation will result in better management of hotel services, benefiting both hotel companies and their customers.

This research uses fuzzy set comparative qualitative analysis (fsQCA), an analysis methodology that has been proven to be adequate to analyse the relationship between an objective variable (NPS) and all the possible combinations of predictors according to the dependent variables of the study (customer satisfaction with reception, cleanliness and room comfort, and gastronomy). The fsQCA methodology manages to identify the possible combinations of these four factors in an optimal way.

The final solution for the month of August 2021 showed us two combinations that increased NPS sufficiently in 95% of the cases with 85% coverage. Specifically, high levels of satisfaction in cleanliness and room comfort and in reception sufficiently increase NPS. This study obtained the same result with high levels of gastronomy quality and service, even if accompanied by low satisfaction in cleanliness and room comfort and reception.

Given that, unlike quantitative estimation techniques, the fsQCA methodology is not symmetrical, it was considered interesting to study which combinations of factors lead to a low NPS. In this case, the combination of low satisfaction in cleanliness and room comfort and high satisfaction in gastronomy and reception leads sufficiently to a low NPS. Likewise, the unique combination of low gastronomy and reception satisfaction leads sufficiently to a low NPS.

Regarding the analysis carried out on the 2020 sample, the final solution also showed two combinations that increase NPS sufficiently in 96% of the cases with 87% coverage. Specifically, high levels of satisfaction in the three study variables sufficiently increase NPS; and this study obtained the same result with high levels of satisfaction in cleanliness and room comfort, even if accompanied by low satisfaction in gastronomy and reception.

Therefore, in August 2020, during the first year of the COVID-19 pandemic, the key factor is cleanliness and room comfort, whereas in August 2021, the key factor is gastronomy.

The 2020 results are in line with the research of Metha et al. [75], as well as that of Yu et al. [76], who identified the perceived hygiene attributes necessary to retain and satisfy customers during the COVID-19 pandemic. Customers typically expressed concern about the hygiene standards of a hotel and the efforts to provide a good experience [76]. During the COVID-19 pandemic, customers rated their satisfaction more highly with Chinese hotels through the website Tripadvisor when they saw that hotels were able to offer cleanliness and health security [77]. Visible hygiene practices directly impact customers' perceptions of cleanliness.

The focus of customers in the same hotels in 2021 appears to be different: in this sample, cleanliness is less of a consideration and gastronomy becomes the key factor to increase the NPS. Zhang and Kim [78] offered different results for a sample of 1493 online reviews about Disneyland Hong Kong between 2017 and 2021, where the quality of the food was negatively related to customer satisfaction.

It is understandable that at the beginning of the pandemic, customers who could afford to travel were more vigilant regarding cleanliness and room comfort, and that this was the decisive factor in boosting the NPS index. An understanding of this can be of use to the successful development of hotel operations, since technology development has facilitated increased communication and dissemination of customer experience [79]. Furthermore, operational failures or negative experiences are quickly exposed on social media. Hotel services focus on intangible elements that generate a lasting emotional impact on consumers, in the knowledge that subsequent reviews and recommendations have a significant impact on future engagement with the business [3].

Therefore, it is vital that the application of satisfaction surveys considers the values and services that are important to customers to ensure that businesses are able to act on them. Various authors have investigated this aspect extensively, using a variety of methods for the selection of items of interest to consumers [80]. For example, Dolnicar and Otter [72] reviewed 21 studies from 1984 to 2000 and were able to summarize the attributes in certain areas of the hotel: image, price/value, the hotel itself, room, services, marketing, food and beverages, others, security, and location. They also pointed out that satisfaction surveys should pay attention to the pre-stay, stay, and post-stay periods. Cleanliness and the location of the hotel were highlighted as the two most significant attributes for customer satisfaction.

Cobanoglu et al. [81] stated that service was the most important factor, followed by price and value, security, additional amenities, technology, room comfort, and food and beverages.

It is evident that, depending on the circumstances, customers may value individual aspects of the hotel differently, and this assessment will also influence their loyalty, expressed through ratios such as the NPS. Taking this into account, it is necessary to understand hotel operation as a living, changing entity, subject to different factors that can affect customer satisfaction. This consideration is vital for successful management in this sector.

In addition to this, the hospitality industry has undoubtedly changed radically due to the impact of COVID-19, which has acted as a BS event for the hotel sector. The perception of customer service has changed over the course of the pandemic. Notably, contact limitation measures such as contactless service may be here to stay, and the impact of these changes on customer satisfaction should be assessed [12].

This study presents some limitations that future research could address. Specifically, when considering the hotel category, NPS and customer satisfaction were analysed in a sample of hotels located in a holiday destination, as opposed to a city-break destination. Furthermore, all six hotels belong to the same hotel company. A direct limitation of this could be that the results cannot be extrapolated to other types of accommodation. Future studies could include city hotels in the sample, and it would also be interesting to use hotels from different companies and locations.

These results offer points of reflection for hotel owners, managers, and those in similar industries, as well as new approaches for scientific research in this area.

**Funding:** This research received no external funding.

**Institutional Review Board Statement:** Not applicable.

**Informed Consent Statement:** Informed consent was obtained from all subjects involved in the study.

**Data Availability Statement:** The data presented in this study are available on request from the corresponding author.

**Conflicts of Interest:** The author declares no conflict of interest.

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
