# Peer review of "Net Promoter Score (NPS) and Customer Satisfaction: Relationship and Efficient Management"

_sustainability, doi:10.3390/su14042011_

Round 1
Reviewer 1 Report
- The abstract is lengthy. Kindly reduce the length by focusing on the salient points.
- NPS is applied everywhere - not only in hotel industry.
- Only six hotels in one location of Spain is situated. Can the results be generalized for hotels in other locations?
- So much description about NPS is not required.
- Why is data based on NPS recorded to determine customer satisfaction.
- How are the dimensions for the analysis decided?
Author Response
I thank you for your detailed comments and suggestions. In light of your comments, I have made several changes, which in my opinion have notably improved the manuscript.
Point 1. The abstract is lengthy. Kindly reduce the length by focusing on the salient points.
Response 1: Thank you for pointing out this issue, which I totally agree with. I have reduced the abstract from 337 to 253 words.
Point 2. NPS is applied everywhere - not only in hotel industry.
Response 2: That is correct. In the paper it does not say that NPS applies only to hotel industry. In fact, it is mentioned that it is used in hospitals and other industries. However, in order to clarify this point for readers, a comment has been included in the Introduction, line 74, to emphasis that “various industries” use NPS.
Point 3. Only six hotels in one location of Spain is situated. Can the results be generalized for hotels in other locations?
Response 3: The results cannot be generalized for other hotels and locations. It does not say so in the paper. In fact, results are related to “the study sample” as it can be read in the Abstract, Results and Conclusions. FsQCA is optimal for analyzing small samples like this one and getting a model for it. It is very usual in the hospitality sector to benchmark or use information from other hotel operations, so this could be one of the contributions of this paper for hospitality executives and owners, as it is expressed in Conclusions.
Point 4. So much description about NPS is not required.
Response 4: Thank you for pointing out this issue. I have removed lines 266 270.
Point 5 and 6. Why is data based on NPS recorded to determine customer satisfaction. How are the dimensions for the analysis decided?
Response 5 and 6: In line 364, Materials and Methods, I explain that the data used for the research comes from a Satisfaction Questionnaire used by an International Hotel Company. The info is managed by ReviewPro. The customer might not know the existence of an NPS index. The customer is answering an online customer satisfaction questionnaire, with questions about the main areas of the hotel Reception, Cleanliness and Room Comfort, Gastronomy, Entertainment and SPA. Entertainment and SPA areas were not included in the research since COVID-19 situation has specially affected these areas, not being in use in some cases. So, the 3 main areas/dimensions used for the research were Reception, Cleanliness and Room Comfort, and Gastronomy.
I thank you again for your detailed comments and suggestions, which have certainly improved the manuscript.

Reviewer 2 Report
The research background/statement of problem is not clear. The issue/problem is not well discussed. gap research between Satisfaction and NPS (loyalty) can be improve and clear . Please mention the similar research works and discuss the existing gap
explanation of the NPS method (208-232) can be transferred to the method
Result: Author should be able to show the NPS calculation table, for example, how many respondents answered the detractors, passive and promoters so that the results of the calculation of the NPS formula can be clearly read by the reader. Please elaborate and explain the gap research with the theory and past references. Moreover, Author should highlight key findings.
In general, the article has enough reference, but it needs to increase by referring to the latest research resources. Strengthen the statements by using more recently references of the past five years.
Discuss solutions, recommendations and limitation in dealing with the issues (gap), controversies, or problems presented.
Author Response
I thank you for your detailed comments and suggestions. In light of your comments, I have made several changes, which in my opinion have notably improved the manuscript.
Point 1. …mention the similar research works and discuss the existing gap.
Response 1: Thank you for pointing out this issue. In fact there is no similar research on this existing gap, so I have add a sentence, line 196, to mention this explicitly: “No similar research has been found in previous studies, so this research adds new insights to the existing gap between NPS and customer satisfaction”.
Point 2. NPS method…
Response 2: NPS method is a very basic calculation. There is no need of Boolean algebra, as used for fsQCA, which is the key issue on this research. I do not have the data of detractors, passive nor promoters, but I could obtain it. As explained in line 360, data has been shared by the hotel company using ReviewPro software. Data used was NPS indexes. Accessing ReviewPro software it is possible to obtain the detail of detractors, passive and promoters for each hotel, but makes no sense to do so for the current research.
Point 3. Discuss solutions, recommendations and limitation in dealing with the issues (gap), controversies, or problems presented.
Response 3: These ideas have been deepened in the paper with the changes made thanks to the comments of the 3 reviewers.
I thank you again for your detailed comments and suggestions, which have certainly improved the manuscript.

Reviewer 3 Report
The article is good but need improvement particularly in literature review which need to be up dated. The author need to specify the contribution of study. Methodology is applicable. Please send to the English editor for proofreading.
Author Response
I thank you for your detailed comments and suggestions. In light of your comments, I have made several changes, which in my opinion have notably improved the manuscript.
Point 1. The article is good but need improvement particularly in literature review which need to be up dated.
Response 1: In order to be more concise and updated, I have reduced from 337 to 253 words the abstract and deleted lines 266 to 270 in the literature review.
Point 2. The author need to specify the contribution of study.
Response 2: I am a single author. I checked previous single author articles and they do not specify the contribution of study (i.e. Kaczmarek, 2022).
Point 3. Please send to the English editor for proofreading.
Response 3: The current manuscript has already been sent to MDPI proofreading.
I thank you again for your detailed comments and suggestions, which have certainly improved the manuscript.
